# The Fission Yeast RNA-Binding Protein Meu5 Is Involved in Outer Forespore Membrane Breakdown during Spore Formation

**DOI:** 10.3390/jof6040284

**Published:** 2020-11-13

**Authors:** Bowen Zhang, Erika Teraguchi, Kazuki Imada, Yuhei O. Tahara, Shuko Nakamura, Makoto Miyata, Satoshi Kagiwada, Taro Nakamura

**Affiliations:** 1Department of Biology, Graduate School of Science, Osaka City University, Sumiyoshi-ku, Osaka 558-8585, Japan; bowenchn1313@gmail.com (B.Z.); tera20eri20@gmail.com (E.T.); imada@chem.suzuka-ct.ac.jp (K.I.); tahara@sci.osaka-cu.ac.jp (Y.O.T.); miyata@sci.osaka-cu.ac.jp (M.M.); 2Department of Chemistry and Biochemistry, National Institute of Technology, Suzuka College, Suzuka 510-0294, Japan; 3The OCU Advanced Research Institute for Natural Science and Technology (OCARINA), Osaka City University, Sumiyoshi-ku, Osaka 558-8585, Japan; 4Department of Biological Sciences, Faculty of Science, Nara Women’s University, Nara 630-8506, Japan; snakamura1030bio@gmail.com (S.N.); kagiwada@cc.nara-wu.ac.jp (S.K.)

**Keywords:** *Schizosaccharomyces pombe*, sporulation, forespore membrane, spore wall, RNA-binding protein, cytochrome *c*

## Abstract

In *Schizosaccharomyces pombe*, the spore wall confers strong resistance against external stress. During meiosis II, the double-layered intracellular forespore membrane (FSM) forms de novo and encapsulates the nucleus. Eventually, the inner FSM layer becomes the plasma membrane of the spore, while the outer layer breaks down. However, the molecular mechanism and biological significance of this membrane breakdown remain unknown. Here, by genetic investigation of an *S. pombe* mutant (E22) with normal prespore formation but abnormal spores, we showed that Meu5, an RNA-binding protein known to bind to and stabilize more than 80 transcripts, is involved in this process. We confirmed that the E22 mutant does not produce Meu5 protein, while overexpression of *meu5^+^* in E22 restores the sporulation defect. Furthermore, electron microscopy revealed that the outer membrane of the FSM persisted in *meu5*∆ spores. Investigation of the target genes of *meu5*^+^ showed that a mutant of *cyc1*^+^ encoding cytochrome *c* also showed a severe defect in outer FSM breakdown. Lastly, we determined that outer FSM breakdown occurs coincident with or after formation of the outermost Isp3 layer of the spore wall. Collectively, our data provide novel insights into the molecular mechanism of spore formation.

## 1. Introduction

Sporulation in the fission yeast *Schizosaccharomyces pombe* is a common cellular process that occurs in response to nutrition starvation, especially nitrogen starvation [1,2]. Sporulation consists of two coordinated processes: meiosis and spore morphogenesis. During meiosis II, a double-layered intracellular membrane called the forespore membrane (FSM), which subsequently becomes the spore plasma membrane, is formed de novo on the cytoplasmic side of the spindle pole body [1]. The FSM then expands by the fusion of membrane vesicles and eventually encapsulates a haploid nucleus generated by meiotic nuclear division, producing the “prespore”, namely the membrane-bound precursor of the spore [1,3,4,5]. Spore wall formation proceeds by the deposition of spore wall materials in the space between the inner and outer FSM of the prespore [1,3]. Eventually, the inner layer of the FSM becomes the spore plasma membrane, while the outer layer of the membrane breaks down, producing a mature spore [1,6]. 

The spore wall confers on the spore strong resistance against various types of stress, such as heat, digestive enzymes, and organic solvents [7,8]. The stress tolerance of a spore is due largely to the robust structure of the spore wall. Formation of the spore wall has been well characterized in the budding yeast *Saccharomyces cerevisiae* [9]. The spore wall of *S. cerevisiae* comprises four distinct layers [10]. The two inner layers are structurally similar to those in the vegetative cell wall, albeit in a reversed order, and consist mainly of mannoproteins and glucans [11]. The third and fourth layers are specific to the spore, and comprise chitosan and a dityrosine-containing polymer, respectively [12,13].

Although formation of the spore wall of *S. pombe* is thought to proceed in a similar way to that of *S. cerevisiae* [1,2], the chemical composition the *S. pombe* spore wall remains unknown. While the glucan layer of the *S. cerevisiae* spore wall is composed of β-glucans, that of *S. pombe* possesses both α- and β-glucans [14]. In addition, the mutants lacking chitin synthase or chitin deacetylase (converts chitin to chitosan) show defect in proper spore wall formation [15,16,17]. The most significant difference in the spore wall structure between these two yeasts is the outermost layer: in *S. cerevisiae*, the outermost layer is composed mainly of dityrosine [12], whereas the *S. pombe* spore is covered by an extraordinarily abundant protein termed Isp3 [18,19]. In both yeasts, the outer layer of the FSM (corresponding to the prospore membrane in *S. cerevisiae*) dissolves during sporulation [1,20], suggesting that breakdown of this membrane contributes to proper spore wall formation. However, the molecular mechanism and physiological significance of outer membrane breakdown remain unknown.

In this study, we present evidence that the RNA-binding protein Meu5 and its target partner cytochrome *c* are responsible for outer FSM breakdown. Moreover, we propose the possibility that breakdown of the *S. pombe* outer FSM occurs after or coincident with the formation of the outermost spore layer of Isp3.

## 2. Materials and Methods 

### 2.1. Yeast Strains, Media, and Culture Conditions

The *S. pombe* strains used in this study are listed in Appendix A. Complete (YE) medium and synthetic (SD, MB, and MM+N) media supplemented with essential nutrients were used for vegetative culture. Malt extract (ME) medium and synthetic sporulation (SSA and MM-N) media supplemented with essential nutrients were used for mating and sporulation [21,22]. *S. pombe* cells were grown at 28 °C or 30 °C and sporulated at 28 °C.

### 2.2. Fluorescence Microscopy

Fluorescence images were captured with a BX53 microscope (Olympus, Tokyo, Japan) and a complementary metal-oxide semiconductor camera (ORCA-Flash4.0, Hamamatsu Photonics, Hamamatsu, Japan) controlled by HC-ImageLive software (Hamamatsu Photonics). Digital images were processed with ImageJ (National Institutes of Health, Bethesda, MD, USA).

### 2.3. Quick-Freeze Deep-Etch Electron Microscopy

*S. pombe* cells were sporulated on ME medium for 2 days. A droplet of sporulating cell suspension, which was washed twice with distilled water, was mixed with mica flake slurry, placed on a 3 × 3 mm piece of rabbit lung slab, and then frozen at liquid helium temperature using a CryoPress (Valiant Instruments, Ellisville, MO, USA). To prepare a replica, the frozen sample was knife-fractured, followed by platinum and carbon rotary-shadowing in a JFDV freeze-etching device (JEOL, Akishima, Japan). The replica was floated on the surface of hydrofluoric acid, cleaned in commercial bleach, rinsed in distilled water, and transferred to a 400-mesh copper grid, as previously described [23,24]. Replicas were viewed under a transmission electron microscope (JEM-1010, JEOL) at 80-kV acceleration voltage. EM images were obtained with a FastScan-F214 (T) charge-coupled device (CCD) camera (TVIPS, Gauting, Germany).

### 2.4. Thin-Section Electron Microscopy

Cells were collected by centrifugation, mounted on formvar (Nisshin EM, Tokyo, Japan) films formed inside a cooper wire loop (8 mm in diameter), and quickly frozen in liquid propane cooled with liquid nitrogen. The frozen cells were transferred to acetone containing 2% osmium tetroxide (EM Japan, Tokyo, Japan) and 0.2% uranyl acetate, and kept at −85 °C. After 2 days, the samples were slowly warmed to 4 °C and kept at room temperature for another 1 hr. After washing with acetone eight times, the samples were gradually substituted with Spurr resin (at 25%, 50%, and 75% in acetone), and finally embedded in 100% Spurr resin. Ultrathin sections were obtained by using an ultramicrotome (EM UC6, Leica, Wetzlar, Germany), stained with lead citrate for 3 min at room temperature, and examined under a transmission electron microscope (JEM-1230, JEOL).

### 2.5. Isolation of meu5^+^

The E22 mutant was transformed with the *S. pombe* genomic library pTN-L1 [4], containing Sau3AI fragments constructed in the multicopy plasmid pAL-KS [25]. Transformants were incubated on SD medium at 28 °C for 4 days and then transferred onto SSA medium. Colonies were exposed to iodine vapor [26]. Iodine staining-positive colonies were removed and inspected for recovery of sporulation after confirmation by fluorescence microscopy. Plasmids were isolated from these candidates and their nucleotide sequences were determined.

### 2.6. Reverse Transcription PCR

Total RNA from sporulating cells was extracted by the glass bead method using ISOGEN reagent (Nippon Gene, Toyama, Japan) and DNase I (Stratagene, La Jolla, CA, USA) treatment. A ReverTra Ace-α-kit (Toyobo, Osaka, Japan) was used for cDNA synthesis. Reverse transcription-PCR (RT-PCR) was performed using two sets of forward and reverse primers (itr1f/itr1r and itr2f/itr2r) (Appendix A).

### 2.7. Western Blotting

BW46 and BW47 cells carrying a single chromosomal copy of *meu5^+^* or the *meu5-E22* gene tagged with three tandem copies of the HA epitope, respectively, were precultured to mid-log phase in MM+N medium and then transferred into MM-N medium to induce sporulation. Samples of the culture were collected at specific intervals, and crude cell extracts were prepared as previously described [27]. Proteins were resolved by SDS-PAGE on a 12% Bis-Tris gel and then transferred onto a polyvinylidene difluoride membrane (Immobilon-P, Millipore, Darmstadt, Germany). Blots were probed with rat anti-HA 3F10 antibody (Roche Diagnostics, Indianapolis, IN, USA) and mouse anti-α-tubulin TAT-1 antibody [28] at a 1:10,000 dilution. Immunoreactive bands were detected by using ECL Select chemiluminescence (GE Healthcare, Little Chalfont, UK) with horseradish peroxidase-conjugated goat anti-rat immunoglobulin G (BioSource International, Camarillo, CA, USA) and sheep anti-mouse IgG (GE Healthcare) at a 1:100,000 dilution.

### 2.8. Construction of the Plasmid for meu5^+^ Overexpression

Plasmid pBW1 was constructed as follows. A *meu5^+^* fragment including the promoter and terminator regions was amplified by PCR using the primers meu5pc.1 and meu5pc.2 (Appendix A). The resulting fragment was digested with SacI and ApaI restriction enzymes, and inserted into the corresponding site of pDblet [29], yielding pBW1.

## 3. Results

### 3.1. The E22 Mutant Shows a Defect in Spore Maturation

In a previous study, we isolated a number of novel sporulation-deficient mutants from mutagenized *S. pombe* cells, in which the FSM was visualized by GFP-tagged Psy1 [30]. To investigate aspects of the sporulation mechanism, here we focused on the E22 mutant, whose asci produced four spores, but the spores were apparently abnormal (Figure 1A). Like other sporulation-deficient mutants, the E22 mutant formed an iodine staining-negative colony on sporulation medium (Figure 1B). Under the differential interference contrast (DIC) microscope, spores from the E22 cells were dull in appearance as compared with spores from wild-type cells (Figure 1A). Mating frequency and progression of meiotic nuclear division were normal in the E22 mutant as judged by the observation of nuclear staining. These data indicate that a late step of spore formation is defective in the E22 mutant. 

In *S. pombe* sporulation, spore wall formation initiates after closure of the FSM [1]. Because the FSM is a double unit membrane, FSMs labelled by GFP-Psy1 are transiently observed as a double ring by fluorescence microscopy, and then the outer ring disappears. Coincidently, the outline of spores becomes clearly visible under a DIC microscope. Hereafter, we refer to spores whose outlines are clearly visible under a DIC microscope as mature spores. 

We evaluated the efficiency of spore maturation in the E22 mutant. At 48 h after the induction of sporulation, most wild-type asci had mature spores. By contrast, only 12% of E22 asci had mature spores (Figure 1A,C). Fluorescence microscopy revealed that most E22 asci included four prespores surrounded by GFP-Psy1, similar to wild-type asci, suggesting that prespore formation proceeds normally in E22 cells (Figure 1D). However, most wild-type spores lost the outer FSM, while the majority of E22 mutant spores retained the outer FSM. To evaluate the efficiency of outer FSM breakdown, we roughly classified the asci into two types: those in which the outer FSM of any of the four spores was observed (type I); and those in which the outer FSM of all four spores disappeared (type II). We then determined the frequency of type II asci (Figure 1E). At 48 h after the induction of sporulation, more than 80% of wild-type asci had completed outer FSM breakdown. In the E22 mutant, by contrast, the abundance of type II asci was less than 20% (Figure 1A,E). Taken together, these data indicate that the E22 mutant is defective in breakdown of the outer FSM. 

### 3.2. The E22 Mutation Is Located in the meu5^+^ Gene Encoding the RNA-Binding Protein Meu5 

The gene responsible for the E22 mutation was identified by transforming the E22 mutant with a genomic library and isolating clones that showed restoration of the sporulation defect (see Materials and Methods). The resulting gene was identical to the previously characterized *meu5^+^/**crp79^+^* gene [31,32], which encodes a protein of 710 amino acids with three RNA recognition motifs. A literature search revealed that *meu5^+^* is upregulated during meiosis [31], while Meu5 binds to and stabilizes the transcripts of more than 80 genes [33]. Hereafter, we refer to the E22 mutant as *meu5-E22*.

Nucleotide sequence analysis demonstrated that the *meu5-E22* mutant harbored a single nucleotide change (from T to G) in the second intron of the *meu5^+^* gene (Figure 2A). The consensus sequence GTANG at the 5′ end of the intron was mutated to GGANG in intron 2 of *meu5-E22*. To detect whether the splicing efficiency of intron 2 was reduced in *meu5-E22*, we conducted reverse transcription PCR analysis with primers encompassing intron 1 and intron 2 (Figure 2A). Total RNA was prepared from sporulating cells of wild type and the *meu5-E22* mutant. As expected, the splicing efficiency of intron 2 but not intron 1 was strikingly reduced in the *meu5-E22* mutant (Figure 2B). Thus, this single nucleotide mutation inhibits pre-mRNA splicing of intron 2 in the *meu5-E22* mutant. 

To examine whether the *meu5-E22* mutant synthesizes Meu5 protein correctly, we constructed strains carrying a single chromosomal copy of either *meu5* or the *meu5-E22* gene tagged with three tandem copies of the HA epitope. In western blot analysis, no Meu5-E22-HA protein band was detected in the *meu5-E22* mutant, whereas a Meu5-HA was detected in wild type (Figure 2D). This result further suggested that the abnormal phenotype of *meu5-**E22* was caused by the absence of Meu5 protein. In fact, the *meu5*∆ deletion strain exhibited essentially the same phenotype as *meu5-E22*, and the outer FSM breakdown defect of both mutants was restored by transformation with a plasmid overexpressing *meu5^+^* (Figure 2E). Considering these results, we concluded that the defect in outer FSM breakdown in the *meu5-**E22* mutant is indeed caused by a defect in Meu5 protein synthesis.

### 3.3. Meu5 Is Dispensable for Initiation of FSM Formation and Expansion of the FSM

As described above, prespores seemed to form normally in the *meu5-E22* mutant (Figure 1A). To confirm this, we observed the initiation of FSM formation and expansion of the FSM in wild-type and *meu5*∆ strains expressing GFP-Psy1 in more detail (Figure 3). Progression of meiosis was monitored by observing the formation and elongation of spindle microtubules. At metaphase II, four semicircle signals of GFP-Psy1 emerged from the vicinity of a nucleus and then elongated to encapsulate the nucleus. Most haploid nuclei produced by meiotic second divisions were encapsulated by the FSM in both wild-type (Figure 3A) and *meu5*∆ mutant cells (Figure 3B), and the processes were indistinguishable. Similar to *meu5-E22*, *meu5*∆ asci had prespores with a double ring of GFP-Psy1. Interestingly, in some of the *meu5* mutant asci, the two outer FSMs of two spores were fused (Appendix A). By contrast, this phenotype was not observed in wild-type asci. Together with the results of a previous study in which *meu5*∆ cells were able to mate and proceed through the meiotic divisions with normal kinetics [33], these data indicate that Meu5 is dispensable for the progression of meiosis and FSM expansion. 

### 3.4. Meu5 Is Involved in Breakdown of the Outer FSM 

We used electron microscopy to investigate the spores in *meu5*∆ in more detail. First, we conducted quick-freeze deep-etch replica electron microscopy [34,35]. In this method, the specimen is frozen in less than a millisecond, and then fractured and exposed to etching and shadowing by platinum, enabling high-contrast images to be obtained [36,37]. As previously reported [38], wild-type spores were observed as a characteristic surface structure from which many protrusions projected outward (Figure 4A,C). By contrast, *meu5*∆ spores exhibited a rather smooth surface (Figure 4B,D).

We also conducted thin-section electron microscopy using the freeze-substitution technique. As shown in Figure 5, both wild-type and *meu5*∆ asci had four spores, but the spore wall in *meu5*∆ was thicker than that in wild type. Consistent with the results of fluorescence microscopy and quick-freeze deep-etch replica electron microscopy, the outer membrane of the FSM persisted in *meu5*∆ spores (Figure 5D,F). By contrast, wild-type spores were surrounded by the electron-dense structure of the Isp3 layer (Figure 5C,E). It should be noted that, in *meu5*∆, the amount of Isp3 is severely reduced [33] and Isp3-GFP did not localize on the surface of spores by fluorescence microscopy (our unpublished data). Taken together, these data further support the notion that Meu5 is involved in breakdown of the outer membrane. 

### 3.5. Cytochrome c Is Involved in Breakdown of the Outer FSM

Meu5 has many targets known to be involved in the progression of meiotic nuclear division and sporulation, including FSM expansion [33,39,40,41]. Nevertheless, *meu5*∆ did not show any defects in these processes. To determine additional genes involved in breakdown of the outer layer, we observed the FSM in several *S. pombe* clones harboring a mutant of the target genes of Meu5. Of the 80+ targets of *meu5^+^* including sporulation-related genes [33], we tested 60 mutants. Notably, *cyc1*∆ cells exhibited the most severe phenotype with regard to breakdown of the outer layer of the FSM (Table 1 and Figure 6). Notably, the role of cytochrome *c*, a highly conserved mitochondrial protein in eukaryotes and most prokaryotes, in sporulation remains unknown. Interestingly, a comparison of DIC images with fluorescence images revealed that spores were clearly visible in most *cyc1*∆ asci even though the outer FSM remained present (Figure 6), suggesting that outer FSM breakdown is independent of the formation of “visible spores” under a DIC microscope.

To examine whether *meu5^+^* and *cyc1^+^* are necessary for the formation of viable spores, we measured the colony-forming ability of their mutant asci (Appendix A). Both *meu5*∆ and *cyc1*∆ asci formed less colonies than wild type, suggesting that these genes are related to forming viable spores.

### 3.6. Localization of Isp3 to the Spore Periphery Precedes Outer Membrane Breakdown

In *S. cerevisiae*, breakdown of the outer membrane occurs after completion of the glucan layer [20]. Lastly, we investigated the timing of outer membrane breakdown in *S. pombe*. Interestingly, fluorescence microscopy revealed that the double ring of the mCherry-Psy1 signal persisted even when the Isp3-GFP ring had formed in *S. pombe* (Figure 7A), suggesting the possibility that breakdown of the outer FSM layer occurs after formation of the Isp3 layer. As described above, the amount of Isp3 is severely reduced in *meu5*∆ [33], therefore we used *cyc1*∆ cells simultaneously expressing mCherry-Psy1 and Isp3-GFP to observe this process. The Isp3-GFP signal almost completely overlapped with the outer ring of FSM in *cyc1*∆ spores (Figure 7B), supporting this possibility.

## 4. Discussion

Similar to nuclear and mitochondrial membranes, the precursor of the spore membrane (the FSM in *S. pombe* and the prospore membrane in *S. cerevisiae*) comprises two units, and the spore wall is formed between these two units. It has been shown that, during or after formation of the glucan layer in *S. cerevisiae*, the outer prospore membrane breaks down and the chitosan layer is subsequently synthesized [20]. Electron microscopy has also revealed that, in *S. pombe*, the outer FSM disappears during spore wall formation [1]. Therefore, it is of interest to know how breakdown of the outer membrane occurs and how it is achieved without damaging the forming spore. In this study, we have provided evidence that, in contrast to *S. cerevisiae,* the outer FSM breaks down after or coincident with the formation of the outermost Isp3 layer in *S. pombe*. Why is the timing of the outer membrane breakdown different between the two yeasts? The glucan layer of the *S. cerevisiae* spore wall is dominated by β-glucans, whereas the *S. pombe* spore wall contains both α- and β-glucans. Furthermore, previous studies have indicated that the chitin/chitosan content is lower in *S. pombe* than in *S. cerevisiae* spore walls, although chitin synthases and chitin deacetylase are important for proper sporulation [15,16,17]. The most significant difference between the spore wall of the two yeasts is the composition of the outermost layer: in *S. cerevisiae*, the major constituent of this “dityrosine layer” is the modified, cross-linked diamino acid *N*-*N*-bisformyl-dityrosine [42,43], whereas *S. pombe* spores are coated by a proteinaceous surface layer comprising mainly Isp3 [18,19]. Thus, we presume that these differences may determine the timing of outer membrane breakdown.

We also identified two genes, *meu5^+^* and *cyc1^+^*, involved in breakdown of the outer membrane. A previous RNA-binding protein immunoprecipitation assay revealed that Meu5 binds to and stabilizes more than 80 target transcripts including the *cyc1*. [33]. Meu5 was originally isolated as an auxiliary mRNA export factor [32], but Meu5 is not directly involved in the export of the *cyc1* mRNA [33]. Therefore, it is more likely that Meu5 is involved in the stabilization rather than the export of the *cyc1* mRNA.

The *meu5*∆ mutant did not show a defect in the progression of meiosis or FSM expansion, but was deficient in spore maturation (Figure 3; [33]). Because Meu5 is an RNA-binding protein, we presumed that targets of Meu5 might be directly involved in outer membrane breakdown. Unexpectedly, however, only *cyc1*∆ cells showed a significant defect in outer membrane breakdown among 60 strains harboring a disruption of a target gene of Meu5 (Table 1). Cytochrome *c* is a small hemeprotein that is loosely associated with the inner membrane of the mitochondrion and is essential for the electron transport system. In *S. pombe*, *cyc1^+^* is not essential for cell viability in normal growth; however, *cyc1*∆ cells show a deficiency in respiration and loss of viability in the stationary phase [44,45]. As a mitochondrial protein involved in the electron transfer system, cytochrome *c* does not seem to be directly involved in membrane breakdown. Indeed, how the mitochondrion is involved in sporulation is little known. In fact, most *cyc1*∆ cells appeared to proceed through normal meiosis and expansion of the FSM. In addition, *cyc1*∆ cells formed visible spores with a clear spore boundary (mature spores), which also suggests that spore maturation is independent of breakdown of the outer FSM (Figure 6 and Figure 7). In *S. cerevisiae*, efficient segregation of mitochondria into spores requires a component of a protein coat found at the leading edge of the prospore membrane [46]; however, it remains to be determined how mitochondrial function is involved in spore formation.

What is the physiological significance of the breakdown of the outer membrane? The release of spores from the ascus was defective or significantly delayed in *meu5*∆ cells [33], suggesting the possibility that outer membrane breakdown is important for spore release. Effective spore release may be important for the spread of spores in the natural environment. Interestingly, mutual adherence between two neighboring spores was occasionally observed in the *meu5* mutant (Appendix A), supporting the possibility. 

We also directly measured the viability of *meu5*∆ and *cyc1*∆ asci. Many of them were able to germinate, although to some extent they were less viable than those of wild type (Appendix A), suggesting the possibility that membrane breakdown is important for proper formation of the spore wall and spore germination. Notably, however, mutation of neither *isp3*^+^ nor *mde10^+^*, both targets of Meu5 and responsible for spore wall formation, affected spore viability [19,38]. Moreover, breakdown of the membrane occurred in these mutants (Table 1 and Figure 6). Given that both Meu5 and cytochrome *c* are involved in various cellular processes, it is unlikely that the reduced spore viability in these mutants directly reflects the ability to form spore walls.

In the present study, we did not identify genes directly involved in breakdown of the outer membrane. In the late stage of sporulation, vacuoles are known to fuse extensively to form a few large membranous compartments that occupy the whole cytoplasm [47]. It is possible that the vacuolar membrane may directly contact the outer FSM and degrade it. Isolation of factors directly involved in breakdown of the outer membrane will be a future challenge.

## Figures and Tables

**Figure 1 jof-06-00284-f001:**
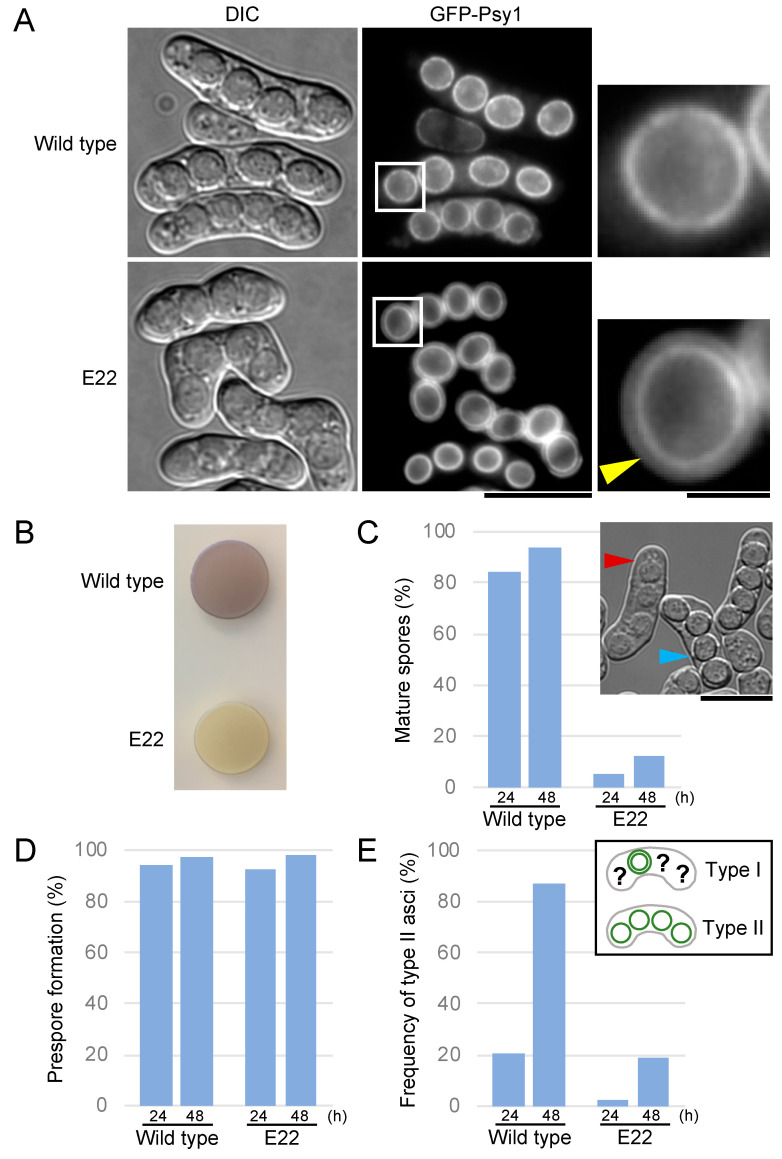
The E22 mutant shows a defect in spore maturation. (**A**) Wild-type (ET1) and E22 mutant cells expressing the forespore membrane (FSM) marker GFP-Psy1 were sporulated on SSA medium for 2 days and analyzed by differential interference contrast (DIC) and fluorescence microscopy. Bar, 10 µm. High-magnification images of the region in the white square are shown on the right. The yellow arrowhead indicates the possible outer layer of the FSM. Bar, 2 µm. (**B**) Wild-type (ET1) and E22 mutant cells were sporulated on malt extract (ME) medium for 2 days and the plate was treated with iodine vapor. (**C**) Sporulation maturation rate of wild type (ET1) and the E22 mutant. Cells were cultured on SSA medium for 24 or 48 h (*n* > 500). In the microscopic image, red and blue arrowheads show immature and mature spores, respectively. Bar, 10 µm. (**D**) Prespore formation is normal in the E22 mutant. Wild-type (ET1) and E22 mutant cells expressing GFP-Psy1 were sporulated on SSA medium for 24 or 48 h and analyzed by fluorescence microscopy (*n* > 500). (**E**) Outer FSM breakdown frequency of wild type (ET1) and the E22 mutant. Cells were cultured on SSA medium for 24 or 48 h and analyzed by fluorescence microscopy (*n* > 200). Question marks represent spores with single or double FSMs.

**Figure 2 jof-06-00284-f002:**
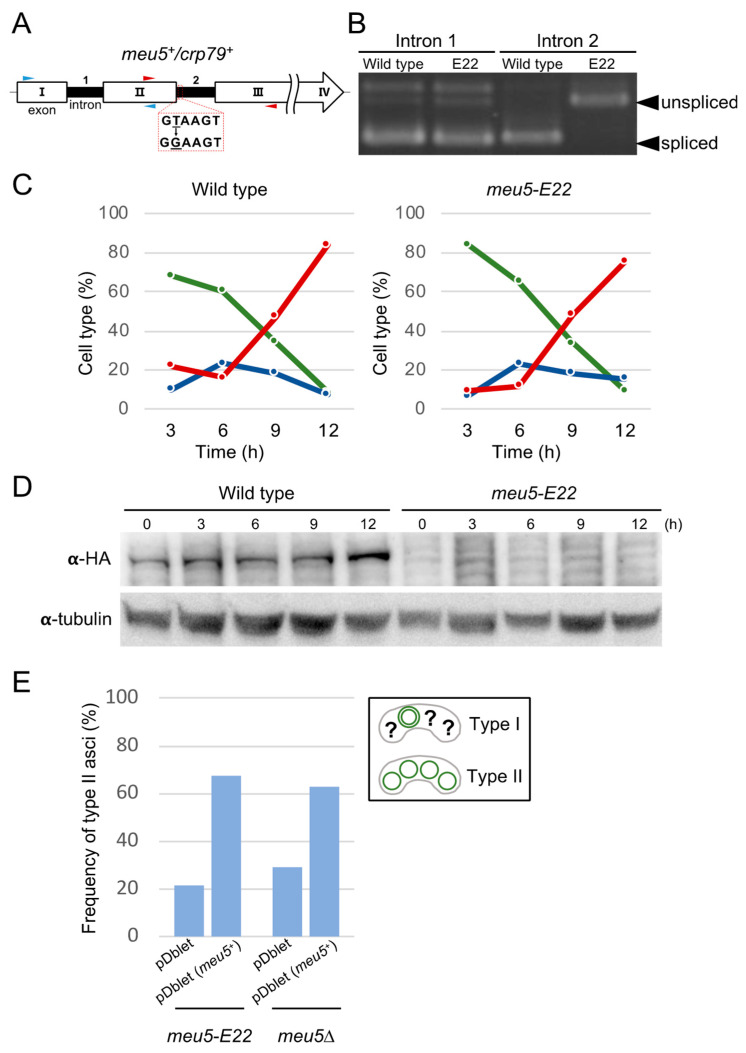
Analysis of the *meu5-E22* mutation. (**A**) Diagram of *meu5^+^* in the E22 mutant. The *meu5-E22* allele carries a single nucleotide change (T to G) in the second intron. The primer sets itr1f/itr1r (blue) and itr2f/itr2r (red) were used to amplify the first and second introns, respectively. (**B**) RT-PCR was used to assess splicing of the first and second introns using the primers shown in panel A. Wild-type (ET1) and *meu5-E22* (E22) cells precultured in MM+N were sporulated in MM-N overnight. (**C**) Kinetics of meiosis in wild type and the *meu5-E22* mutant. Wild-type (BW46) and *meu5-E22* (BW47) cells precultured in MM+N were incubated in MM-N. A portion of the culture was stained with DAPI. Meiotic cells were classified by the number of nuclei per cell (*n* > 200). The figure is based on one of three independent experiments with similar results. Green, mononucleate; blue, binucleate; red, tri- or tetranucleate cells. (**D**) Expression of Meu5 in the *meu5-E22* mutant. Wild-type (BW46) and *meu5-E22* (BW47) cells precultured overnight in MM+N were incubated in MM-N. Protein extracts were subjected to western blot analysis with a rat anti-HA antibody and a mouse anti-α-tubulin antibody as a loading control. (**E**) Complementation of *meu5-E22* and *meu5*∆ by overexpression of Meu5. *meu5-E22* (E22) and *meu5*∆ (BW41) cells were transformed with the multicopy plasmid pDblet [29] carrying *meu5^+^* (pBW1). Transformants were sporulated on SSA medium for 2 days (*n* > 200).

**Figure 3 jof-06-00284-f003:**
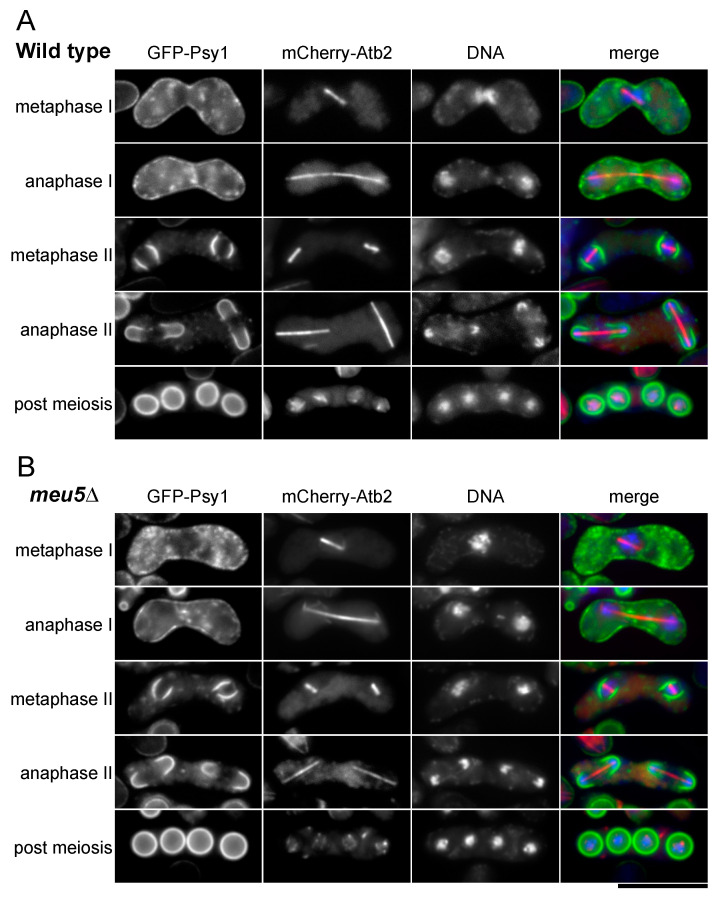
Normal assembly of the FSM in the *meu5*∆ mutant. (**A**) Wild-type (KI173) and (**B**) *meu5*∆ (BW162) strains expressing GFP-Psy1 and mCherry-Atb2 (α-tubulin) were sporulated on SSA medium for 1 day. Chromosomal DNA was stained with Hoechst 33342 and observed by fluorescence microscopy. GFP-Psy1 (green), mCherry-Atb2 (red), and Hoechst 33342 (blue) are overlaid in the merge images. Bar, 10 µm.

**Figure 4 jof-06-00284-f004:**
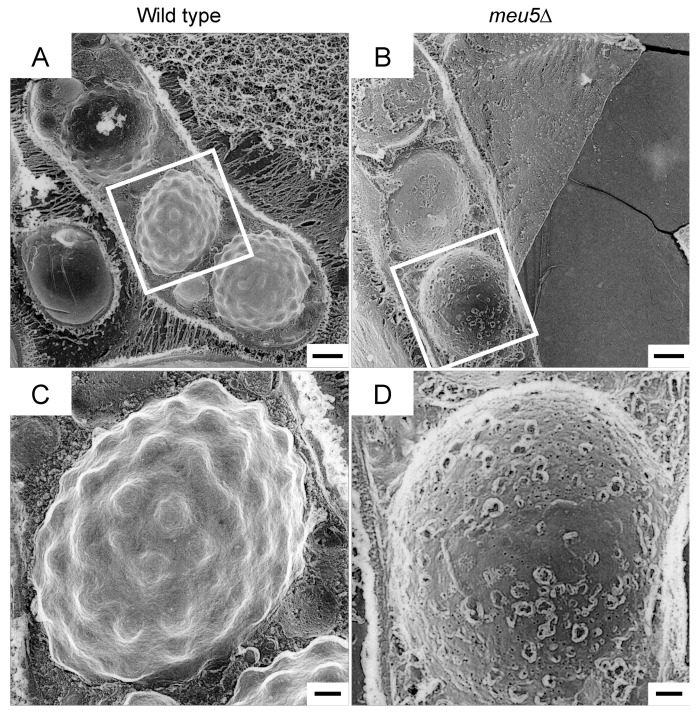
Quick-freeze deep-etch replica electron microscopic images of wild-type and *meu5*∆ spores. Wild-type (KI36) and *meu5*∆ (STA71) cells were sporulated on ME medium for 2 days. (**A**,**C**) Wild type; (**B**,**D**) *meu5*∆. (**C**,**D**) are high-magnification images of the regions in the white square of (**A**) and (**B**), respectively. Bar, 1 µm (**A**,**B**); 0.3 µm (**C**,**D**).

**Figure 5 jof-06-00284-f005:**
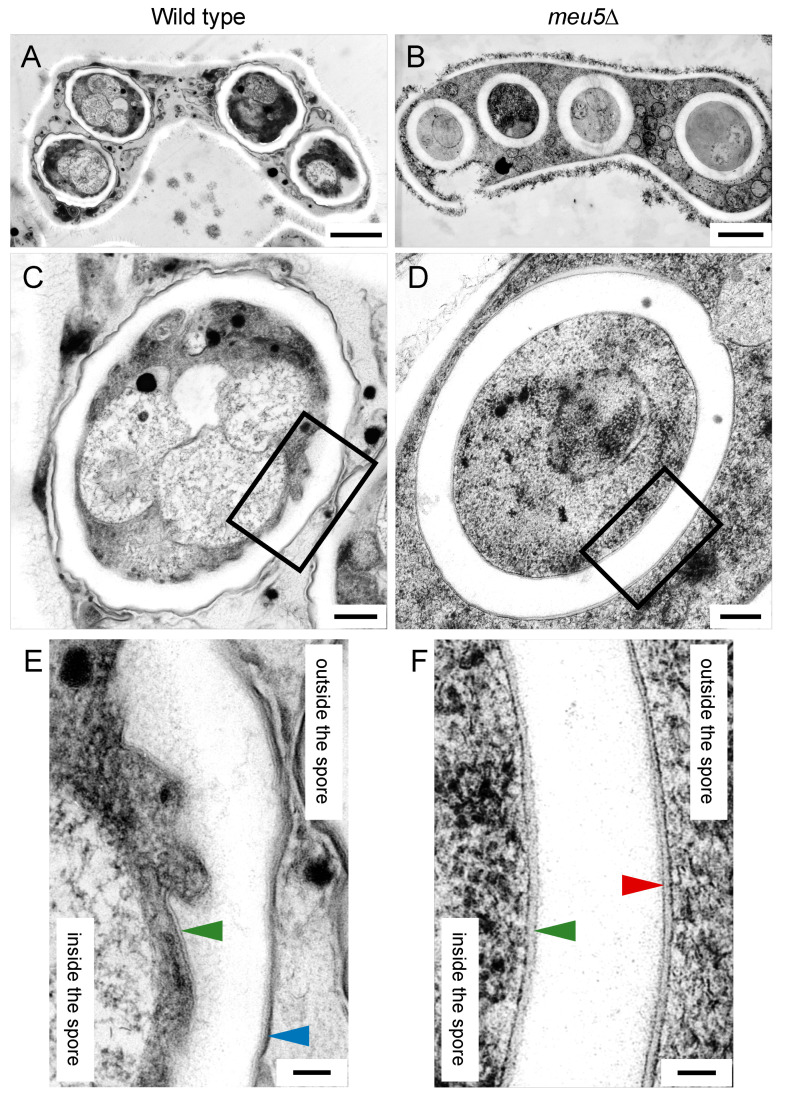
Thin-section electron microscopic images of wild-type and *meu5*∆ spores. Wild-type (YN68) and *meu5*∆ (BW84) cells were sporulated on SSA for 2 days and observed by electron microscopy. (**A**,**C**,**E**) Wild type; (**B**,**D**,**F**) *meu5*∆. (**E**,**F**) are magnified images of the boxed regions in (**C**,**D**), respectively. Green and red arrowheads indicate the inner and outer FSM, respectively. The predicted Isp3 layer is indicated by the blue arrowhead. Bar, 1 µm (**A**,**B**); 200 nm (**C**,**D**); 50 nm (**E**,**F**).

**Figure 6 jof-06-00284-f006:**
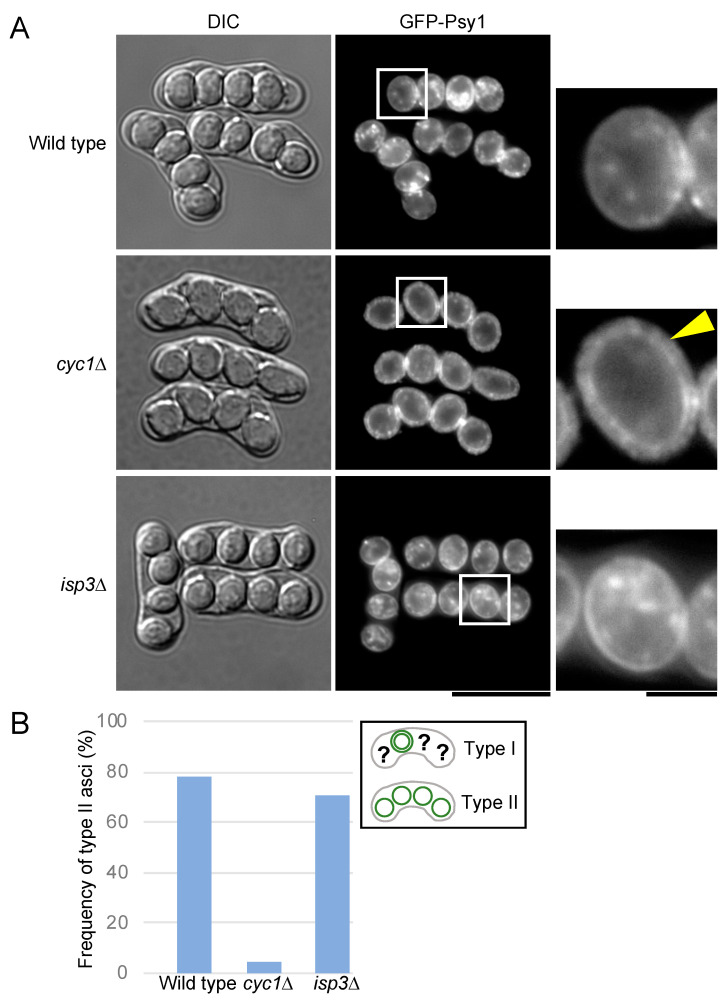
Outer FSM breakdown of *cyc1*∆ and *isp3*∆. (**A**) Wild-type (ET1), *cyc1*∆ (BW22), and *isp3*∆ (BW144) cells were sporulated on SSA medium for 2 days and analyzed by DIC and fluorescence microscopy. Bar, 10 µm. High-magnification images of the regions in the white squares are also shown. The yellow arrowhead indicates the possible outer layer of the FSM. Bar, 2 µm. (**B**) Frequency of outer FSM breakdown in wild-type (ET1), *cyc1*∆ (BW22), and *isp3*∆ (BW144) cells (*n* > 200).

**Figure 7 jof-06-00284-f007:**
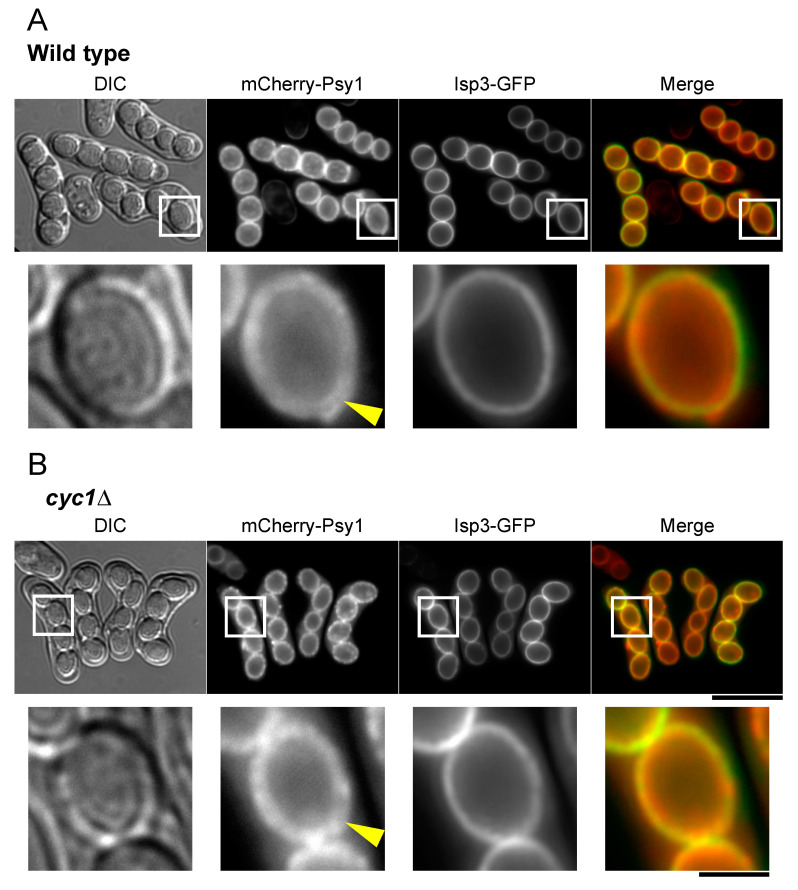
Isp3-GFP localizes to the spore periphery prior to disappearance of the outer FSM. (**A**) Wild-type (BW40) and (**B**) *cyc1*∆ (BW303) cells expressing mCherry-Psy1 and Isp3-GFP were sporulated on SSA medium for 1 day and analyzed by fluorescence microscopy. mCherry-Psy1 (red) and Isp3-GFP (green) are overlaid in the merge images. Bar, 10 µm. High-magnification images of the regions in the white squares are shown on the right. The yellow arrowhead indicates the possible outer layer of the FSM. Bar, 2 µm.

**Table 1 jof-06-00284-t001:** Screening results of *meu5^+^* targets.

Gene	Description (Pombase)	Frequency of Type II Asci (%)
*cyc1*	cytochrome *c*	4.3
*are2*	acyl-CoA-sterol acyltransferase Are2	26.8
*pmp3*	plasma membrane proteolipid Pmp3 (predicted)	27.9
SPBPB2B2.07c	*S. pombe* specific DUF999 protein family 7	29.8
*mug129*	*Schizosaccharomyces* specific protein Mug129	42.1
agn2	glucan endo-1,3-alpha-glucosidase Agn2	43.8
*mug63*	TLDc domain protein 1, implicated in response to oxidative stress	43.8
*atg24 (snx4)*	autophagy associated PX/BAR domain sorting nexin Atg24	44.1
*ctl1*	choline transporter-like, implicated in autophagy Ctl1	45.5
*hhp1*	serine/threonine protein kinase Hhp1	47.5
*pmc1*	vacuolar calcium transporting P-type ATPase P2 type, Pmc1	52.2
*fzr2*	meiotic fizzy-related APC coactivator Fzr2 (predicted)	52.7
*ubp4*	ubiquitin C-terminal hydrolase Ubp4	52.8
*fis1*	mitochondrial fission protein Fis1 (predicted)	53.0
*dsk1*	SR protein-specific kinase Dsk1	54.6
SPAC750.06c	*S. pombe* specific DUF999 protein family 4	56.4
*uvi15*	tail anchored plasma membrane protein Uvi15	57.8
*pcy1 (pct1)*	choline phosphate cytidylyltransferase Pcy1 (predicted)	59.1
*meu6*	pleckstrin homology domain protein Meu6	59.5
*nep2 (mug120)*	NEDD8 protease Nep2	62.6
*omh4*	alpha-1,2-mannosyltransferase Omh4 (predicted)	62.8
*mug58*	GLYK family kinase of unknown specificity, implicated in nucleotide metabolism (predicted)	63.6
*gsk31*	serine/threonine protein kinase Gsk31 (predicted)	66.7
*gor1*	glyoxylate reductase (predicted)	66.7
SPBC1348.01	*S. pombe* specific DUF999 protein family 5	67.0
*isp3 (meu4)*	spore wall structural constituent Isp3	70.7
*sdu1 (mug67, hag1)*	PPPDE peptidase family deubiquitinase/desumoylase Sdu1 (predicted)	71.1
*psk1*	ribosomal protein S6 kinase Psk1	71.4
*elo1*	fatty acid elongase Elo1	72.0
*idn1*	gluconokinase	73.5
*ryh1 (hos1, sat7)*	GTPase Ryh1	73.7
*gma12*	alpha-1,2-galactosyltransferase Gma12	75.2
*git5 (gpb1)*	heterotrimeric G protein beta (WD repeat) subunit Git5	77.0
*mug111*	major facilitator family transmembrane transporter Mug111 (predicted)	77.3
SPAC212.04c	*S. pombe* specific DUF999 family protein 1	77.3
*meu7 (aah4)*	alpha-amylase homolog Aah4	77.5
SPAC212.01c	*S. pombe* specific DUF999 family protein 2	78.3
*pmp31 (mug75)*	plasma membrane proteolipid Pmp31	79.0
*mug110*	*Schizosaccharomyces* specific protein Mug110	79.2
*mde10 (mug139)*	spore wall assembly ADAM family peptidase Mde10	79.2
*mug86*	plasma membrane acetate transmembrane transporter (predicted)	79.6
*meu31*	*Schizosaccharomyces* specific protein Meu31	80.0
*pet2*	Golgi phosphoenolpyruvate transmembrane transporter Pet2	80.4
*pso2 (snm1)*	DNA 5’ exonuclease (predicted)	80.6
*fbp1*	fructose-1,6-bisphosphatase Fbp1	82.5
*meu34 (mug145)*	ubiquitin-protein ligase E3 Meu34, human RNF13 family homolog, unknown biological role (predicted)	83.3
SPAC4F10.16c	plasma membrane phospholipid-translocating ATPase complex, ATPase subunit (predicted)	84.5
*rit1*	initiator methionine tRNA 2’-O-ribosyl phosphate transferase (predicted)	85.8
*gld1*	mitochondrial glycerol dehydrogenase Gld1	86.1
SPCC1739.08c	short chain dehydrogenase (predicted)	88.2
*erf2 (mug142)*	palmitoyltransferase Erf2	89.3
*mug109*	Rab GTPase binding protein upregulated in meiosis II (predicted)	91.6
*mde5 (meu30)*	alpha-amylase homolog Mde5	92.4
*pdh1*	Golgi to ER retrograde transport protein (predicted)	93.3
*eng2*	cell wall and ascospore endo-1,3-beta-glucanase Eng2	93.6
*mpf1*	meiotic pumilio family RNA-binding protein Mpf1 (predicted)	94.1
*mok14*	alpha-1,4-glucan synthase Mok14	99.0
*mug113*	GIY-YIGT nuclease superfamily protein	99.0
*ndk1*	nucleoside diphosphate kinase Ndk1	99.0
SPAC977.06	*S. pombe* specific DUF999 family protein 3	unable to sporulate

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
