# Peer review of "The Fission Yeast RNA-Binding Protein Meu5 Is Involved in Outer Forespore Membrane Breakdown during Spore Formation"

_jof, 2020, doi:10.3390/jof6040284_

Round 1
Reviewer 1 Report
In this paper, the authors have cloned the Meu5 gene from a mutant E22 with a general defect in spore wall maturation. They show by electron microscopy that the outer membrane of the forespore double membrane persisted in the meu5Δ spores compared to the wild type spores. Moreover, by looking at the target genes of meu5+ they found that the cyc1Δ mutant cells also showed a severe defect in outer FSM breakdown. However they have not gone further to demonstrate what is the relationship among Meu5 and Cyc1 and whether the defect in FSM breakdown in meu5Δ mutants is due to defect in the function of cyc1+ (cytochrome c). Overall, I found the manuscript easy to read. The studies are logical and appropriately controlled. Addressing my following concerns may improve the manuscript.
Mayor questions
1) In the introduction, the authors relay too much in S. cerevisiae to explain spore cell wall structure while ignoring the “state of the art” of the spore wall in S. pombe….Some more detail in what is known about the spore wall in fission yeast will improve the manuscript.
2) It is not clear to me how was isolated the meu5+ gene. …..
In the material and methods section, line 102-107…”The E22 mutant was transformed with the S. pombe genomic library pTN-L1 [4]. Transformants were incubated on SD medium at 28ºC for 4 days and then transferred onto SSA medium. Colonies were exposed to iodine vapor [21]. Iodine staining-positive colonies were removed and inspected for recovery of sporulation after confirmation by fluorescence microscopy”.
Is this because the colonies of the E22 mutants are iodine negative, or because the authors have had to analyze one by one all the transformed clones able to sporulate by fluorescence microscopy. If this is the case, it will be informative to know how many clones had to be analyzed before they could restore the phenotype of the E22 mutant.
3) I understand that Meu5 is involved in breakdown of the outer FSM, but I am not totally convinced that the Isp3 abundant layer does not surround the meu5Δ spores (as opposed to the wild type spores).
In that sense I found misleading the interpretation of the results of figure 5 compared with those of figure 7.
In figure 5….and lines 248-250 is written, “the outer membrane of the FSM persisted in meu5Δ spores (Figure 5D and F). By contrast, wild-type spores were surrounded by the electron-dense structure of the Isp3 layer (Figure 5C and E)”….suggesting that the meu5Δ spores are not surrounded by the Isp3 layer.
In Figure 7 and lines 284-288….”fluorescence microscopy revealed that the double ring of the mCherry-Psy1 signal persisted even when the Isp3-GFP ring had formed in S. pombe (Figure 7), suggesting the possibility that breakdown of the outer FSM layer occurs after formation of the Isp3 layer. Here clearly the cyc1Δ spores that conserved the outer FSM also show Isp3-GFP staining. The question is are or aren´t the meu5Δ spores positive for Isp3-GFP, in which case figure 5 might has been miss interpreted.
4) Is Meu5 important for the formation of functional spores? Are the meu5Δ spores more sensitive to glusulase or ethanol as happens with the isp3Δ spores?
5) It is shown in the paper that one of the targets of Meu5 is the alpha-1,4-glucan synthase Mok14 (Table 1). Since Mok14p is required for the synthesis of the iodine-reactive polymer characteristic of S. pombe spores (doi: 10.1111/j.1365-2958.2005.04995), it will be informative to determine whether the meu5Δ spores are iodine positive or not.
6) To understand the relationship between Meu5p and Cyc1p would improve the manuscript. Is Meu5/ involved in Cyc1 mRNA export or stabilization? What is the behavior of the double mutant meu5Δcyc1Δ regarding outer FSM breakdown.
Reviewer 2 Report
This very nice study establishes a role for the RNA-binding protein Meu5 and cytochrome c (encoded by cyc1) in the degradation of the outer layer of the forespore membrane (FSM). The FSM is a double membrane structure that important in the formation of spores, as the inner layer becomes the plasma membrane of the newly formed haploid spore. How the outer lipid bilayer is broken down is not understood and the work in this paper reveals important insights regarding this process.
Specifically, the authors undertake a genetic screen and discover that meu5 plays an important role in spore maturation and the removal of the outer lipid bilayer of the FSM. This defect is shown using both the point mutation allele from the genetic screen (which is mapped to the second intron) and using a deletion allele; the phenotype and can be rescued by overproducing meu5+ in the mutant strain backgrounds. Previous studies had identified targets of meu5; the authors examine many of these targets and discover a previously unknown role of cytochrome c in the breakdown of the outer layer of the FSM. The studies in this paper are well-done and include some beautiful electron microscopic images (both TEM and quick freeze deep-etch replica EM images) that corroborate the fluorescence microscopy phenotypes for these mutant alleles.
The main concerns I have about this manuscript relate to the Supplemental Figures. Currently, none of the data is presented in the Results section, where they should be mentioned. Instead, this data only gets mentioned in the Discussion. Both Supplemental Figures provide intriguing results related to the paper, although neither are key to the important findings of this manuscript. Thus, the data could either could be left in the manuscript and mentioned in the Results section (either as Supplemental Data or as part of a main Figure) or removed from the manuscript without affecting the current set of results.
For Figure S1, I think that fusion of the outer layer of the FSM seen in the meu5 mutants in using both EM and fluorescence microscopy is very interesting, although it would be nice to see some quantitation of this phenotype using fluorescence microscopy, to get a feel of whether this phenotype is always seen in most cells or whether this is something that is seen in a low percentage of meu5 cells – and whether you ever see this phenotype in wild type cells (and perhaps in cyc1 cells). Alternatively, although intriguing, this data is not necessary for this study, and without knowing whether this phenotype is just found at a low frequency in odd cells in the culture or whether this is a robust phenotype, it might be better to leave this out.
The data in Figure S2 are intriguing because it suggests an important biological role for the outer FSM layer. For this data, it would be nice to have biological replicates of the spore germination phenotype, particularly as it seems a little unexpected that the meu5 knockout has a less severe phenotype compared to the cyc1 knockout, since the cyc1 mutants could be thought of has having a less severe sporulation phenotype (since the Isp3 spore wall protein seems more normally localized compared to a meu5 mutant).
